EMBO
Molecular Medicine

# Akt inhibition improves long-term tumour control following radiotherapy by altering the microenvironment

Emma J Searle[1,2,3], Brian A Telfer[1], Debayan Mukherjee[2,3], Duncan M Forster[4], Barry R Davies[5], Kaye J Williams[1,6,7,†], Ian J Stratford[1,†] & Tim M Illidge[2,3,†,*]

## Abstract

Radiotherapy is an important anti-cancer treatment, but tumour recurrence remains a significant clinical problem. In an effort to improve outcomes further, targeted anti-cancer drugs are being tested in combination with radiotherapy. Here, we have studied the effects of Akt inhibition with AZD5363. AZD5363 administered as an adjuvant after radiotherapy to FaDu and PE/CA PJ34 tumours leads to long-term tumour control, which appears to be secondary to effects on the irradiated tumour microenvironment. AZD5363 reduces the downstream effectors VEGF and HIF-1α, but has no effect on tumour vascularity or oxygenation, or on tumour control, when administered prior to radiotherapy. In contrast, AZD5363 given after radiotherapy is associated with marked reductions in tumour vascular density, a decrease in the influx of CD11b[+] myeloid cells and a failure of tumour regrowth. In addition, AZD5363 is shown to inhibit the proportion of proliferating tumour vascular endothelial cells *in vivo*, which may contribute to improved tumour control with adjuvant treatment. These new insights provide promise to improve outcomes with the addition of AZD5363 as an adjuvant therapy following radiotherapy.

**Keywords** Akt; microenvironment; radiotherapy
**Subject Categories** Cancer; Pharmacology & Drug Discovery

## Introduction

Radiotherapy (RT) plays an important part in approximately 40% of all cancer cures (Price *et al*, 2008); however, relapses after RT are common and present an ongoing clinical challenge to further improve outcomes. An increased understanding of potentially important molecular targets and signalling pathways involved in driving cancer has led to the development of novel therapeutics, which can be combined with RT to potentially improve tumour control and outcomes further (Bernier *et al*, 2004). One potential target is Akt, which is at the centre of the phosphatidylinositol-4,5-bisphosphate 3-kinase (PI3K)/Akt pathway, known to have diverse cellular functions including those of survival, growth, proliferation, angiogenesis, glucose uptake and cellular metabolism (Cantley, 2002). Activation of the PI3K/Akt pathway is a prominent feature of many human cancers, and overexpression of the phosphorylated form of Akt has been shown to be associated with radioresistance and local recurrence (Gupta *et al*, 2002).

Several potential mechanisms have been described by which Akt is involved in the response of tumours to RT suggesting the combination of RT with Akt inhibition may be an important therapeutic opportunity (Kim *et al*, 2006; Bussink *et al*, 2008; Schuurbiers *et al*, 2009). It has been demonstrated that inhibitors of the Akt pathway can increase the intrinsic radiosensitivity of tumour cells (Kim *et al*, 2005). Akt is also known to impact on the tumour microenvironment and to induce hypoxia-inducible factor 1 (HIF-1) and vascular endothelial growth factor (VEGF) expression, which in turn can affect the tumour vasculature and tumour response to RT (Kao *et al*, 2007; Jiang & Liu, 2009; Fokas *et al*, 2012).

A number of potent inhibitors of Akt have recently entered clinical development. One such compound, AZD5363, is an orally bioavailable, adenosine triphosphate (ATP) competitive inhibitor of all three isoforms of Akt. AZD5363 is an attractive candidate as an adjuvant to RT as the drug has successfully completed phase I trial investigation with an acceptable safety and tolerability profile (Banerji *et al*, 2013). In this study, we have evaluated whether AZD5363 in combination with RT could improve tumour control.

1 Division of Pharmacy and Optometry, School of Health Sciences, University of Manchester, Manchester, UK
2 Division of Cancer Sciences, School of Medical Sciences, University of Manchester, Manchester, UK
3 Christie Hospital, Manchester Academic Health Sciences Centre, University of Manchester, Manchester, UK
4 Division of Informatics, Imaging & Data Sciences, School of Health Sciences, University of Manchester, Manchester, UK
5 IMED Oncology Biotech Unit, AstraZeneca, Cambridge, UK
6 CRUK-EPSRC Cancer Imaging Centre in Cambridge and Manchester, Cambridge, UK
7 CRUK-EPSRC Cancer Imaging Centre in Cambridge and Manchester, Manchester, UK
*Corresponding author. Tel: +44 161 306 3246; E-mail: tmi@manchester.ac.uk
†These authors contributed equally to this work

   

We have investigated the potential of AZD5363 to radiosensitise cancer cells *in vitro* and increase tumour control after RT *in vivo* in two tumour models of head and neck cancer (FaDu and PE/CA PJ34). This has included establishing the optimal scheduling of AZD5363 and RT in order to maximise the efficacy of the combination.

We have further investigated mechanistically the effects of AZD5363 in combination with RT on the tumour microenvironment. Specifically, we have studied the effect of AZD5363 on the levels of pro-angiogenic proteins VEGF and hypoxia-inducible factor 1-alpha (HIF-1α) and the subsequent impact of any alteration on tumour vascularity and oxygenation. In addition, we have considered the potential effect of AZD5363 on tumour vasculogenesis after RT and on vascular endothelial cell radiosensitivity and proliferation.

These data collectively provide a strong case for the continued development of AZD5363 as a potential adjuvant to improve outcomes after RT. The study also provides potentially important proof of principle scheduling data to inform early phase clinical trial design.

## Results

### AZD5363 inhibition does not enhance the radiosensitivity of tumour cells *in vitro*

It is known that Akt inhibition can alter the intrinsic radiosensitivity of tumour cells (Kim *et al*, 2005); therefore, we evaluated the effect of AZD5363 on a variety of cell lines *in vitro*. AZD5363 is a potent inhibitor of Akt and led to reductions in the levels of phosphorylated Akt substrate glycogen synthase kinase 3 beta (pGSK3β) and the phosphorylated downstream pathway S6 ribosomal protein (pS6) detectable by Western blot in all cell lines (Fig 1A, and Appendix Fig S1A and B). As anticipated with an ATP-competitive inhibitor of Akt, hyperphosphorylation of Akt (pAkt) was detected, and in addition, significant reductions in phosphorylated proline-rich Akt substrate of 40 kDa (pPRAS40) were measurable by enzyme-linked immunosorbent assay (ELISA) with doses of 1 μM and 10 μM AZD5363 (Fig 1B, and Appendix Fig S1C and D).

The effect of AZD5363 on cell viability as measured by the MTT assay was highly variable between cell lines (Fig 1C and Appendix Fig S2A–C). Four cell lines (C33a, ME180, PE/CA PJ34 and MCF-7) were sensitive to AZD5363 (as defined as a > 50% reduction in viability with a dose of < 3 μM). Cal-27, Detroit-562 and HT3 cells displayed moderate sensitivity (> 50% reduction in viability with a dose of 3–10 μM). The remaining cell lines (FaDu, PE/CA PJ15, RPMI 2650, CaSki, SiHa, T47D, A549 and Calu-6) were insensitive to AZD5363 (50% reduction in viability with a dose > 10 μM).

Clonogenic assays were used to assess the effect of AZD5363 on the radiosensitivity of tumour cells *in vitro*. The 15 cell lines (Appendix Table S1) were treated with various concentrations of AZD5363 for 2 h before and 24 h after a single dose of RT. Results showed that AZD5363 did not enhance the radiosensitivity of any of the 15 cell lines investigated (Fig 1D and E, and Appendix Fig S3A–I). These experiments were repeated in FaDu cells using a variety of dosing schedules: (i) AZD5363 applied for 2 h before IR only; (ii) AZD5363 applied for 2 h before RT and continuously thereafter

during colony formation; (iii) AZD5363 applied 4 h post-RT, for 24 h; (iv) AZD5363 applied for 2 h before and for 24 h after RT. None of these sequences enhanced the radiosensitivity of FaDu cells (Fig 1F).

An investigation of the potential effect of AZD5363 on the progression of FaDu cells through the cell cycle, both alone and for 24 h after 4 Gy RT, was conducted. This showed no changes when AZD5363-treated cells were compared to untreated (Appendix Fig S3J), with or without irradiation. However, confirmation that AZD5363 inhibited the downstream Akt pathway (pGSK3β) in FaDu cells after RT was provided by Western blot (Fig 1G).

### AZD5363 following RT results in improved tumour control *in vivo*

Two cancer cell types with differing sensitivity to AZD5363 were selected for further evaluation *in vivo* in combination studies to assess the efficacy of combining AZD5363 with RT. Initially, mice bearing FaDu tumours measuring 100 mm³ were treated with one of three twice-daily drug treatment schedules (neo-adjuvant, continuous or adjuvant), in addition to a single 6 Gy dose of RT (see outline in Fig 2A). Neither 6 Gy RT, nor AZD5363 alone, significantly increased the length of time before tumour control was lost (tumour volume reached 700 mm³), when compared to the vehicle treatment group (median time 30 days vs. 17 days ($P = 0.07$) and 24 days vs. 17 days ($P = 0.225$), respectively; Fig 2B) or decreased tumour growth as assessed by group mean tumour volumes (MTVs) ($P = 0.542$ and $P = 0.871$, respectively; Fig 2C). In contrast, AZD5363 substantially improved tumour control when administered following RT to 81 days, compared with 30 days in the RT alone group ($P = 0.005$; Fig 2B). MTVs were significantly smaller in the adjuvant group than the RT control ($P = 0.002$; Fig 2D). Continuous treatment led to tumour control of 35 days ($P = 0.024$; Fig 2B), which was significantly less than when AZD5363 was given after RT ($P = 0.038$). Administration of AZD5363 prior to RT did not significantly alter tumour control compared to RT alone (23 days vs. 30 days; Fig 2B), nor tumour growth (Fig 2F). At the end of the experiment, (day 100), 50% of those treated with AZD5363 after RT achieved long-term tumour control having no measurable tumour mass. No erythaema or ulceration was observed over the tumour site with any treatment protocol, and mean group mouse weight did not drop significantly throughout (Appendix Fig S4A).

In order to confirm the validity of this observation, this experiment was undertaken in another head and neck human cancer xenograft model. The PE/CA PJ34 model was selected as AZD5363 had been shown to have direct cytotoxicity to this cell line, in contrast to FaDu cells, which we had already demonstrated are insensitive to AZD5363 monotherapy. Subcutaneous PE/CA PJ34 tumours were established in nude mice, and the mice were then randomised into neo-adjuvant, continuous and adjuvant groups (Fig 2A) along with appropriate AZD5363 alone, RT alone and vehicle alone control groups. These experiments confirmed that AZD5363 delivered as an adjuvant after RT, led to significantly improved tumour control compared to RT alone; median time 84.5 days vs. 29 days, $P = 0.044$; Fig 3A). Tumour MTV was also significantly reduced in the adjuvant group compared to RT alone ($P = 0.043$; Fig 3D). No significant effect on MTV was observed from either RT or AZD5363 alone ($P = 0.243$ and $P = 0.2201$, respectively), nor was an effect on tumour control

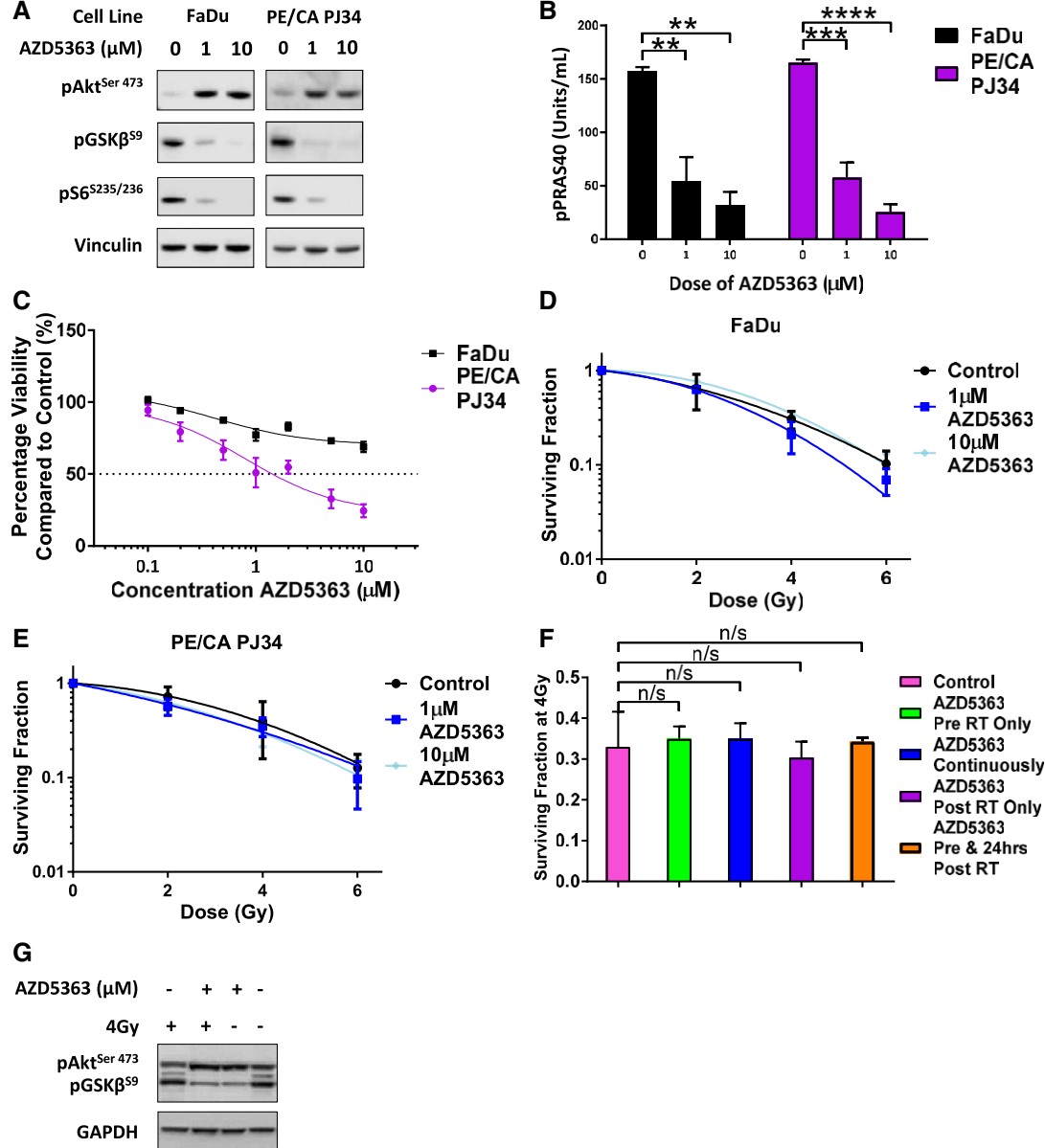

**Figure 1. Akt inhibition with AZD5363 does not enhance the radiosensitivity of tumour cells *in vitro*.**

A, B   Cells were treated with 1 μM or 10 μM AZD5363 for 2 h before lysis on ice. (A) Western blots were performed with antibodies detecting pAkt, pGSK3β, pS6 and the housekeeping protein vinculin. A blot representative of three independent experiments is shown in the panel (*n* = 3 experiments). (B) ELISA was used to measure levels of pPRAS40 (*n* = 3 experiments).

C   MTT assay of cells treated with 1–10 μM AZD5363 for 96 h (*n* = 3 experiments).

D, E   Clonogenic assay of FaDu (D) and PE/CA PJ34 (E) cells treated with 1 μM or 10 μM AZD5363 for 2 h before, and 24 h after a single dose of RT (2, 4 or 6 Gy) (*n* = 3 experiments).

F   Surviving fraction of FaDu cells after a single 4 Gy dose of RT combined with varying schedules of 1 μM AZD5363 (*n* = 3 experiments).

G   Cells were treated with 1 μM AZD5363 for 2 h before and 24 h after a single 4 Gy dose of RT before lysis on ice. Western blots were performed with antibodies detecting pAkt, pGSK3β and the housekeeping protein GAPDH. A blot representative of three independent experiments is shown in the panel (*n* = 3 experiments).

Data information: In (B–F) data are presented as mean ± SEM; n/s *P* > 0.05, \*\**P* < 0.01, \*\*\**P* < 0.001, \*\*\*\**P* < 0.0001. Statistical test in (B) and (F) is one-way ANOVA with Dunnett's *post hoc* test (adjusted *P* = <0.0001–0.005 and *P* = 0.993–0.998, respectively). In (D) and (E) data are fitted to the linear quadratic model. In (C) data are fitted to a dose-response curve.

demonstrated (*P* = 0.069 and *P* = 0.264, respectively). Long-term tumour control with no measurable tumour mass was achieved in 50% (4/8) of adjuvant-treated mice. In the group of mice receiving

the combination of continuous AZD5363 and RT, tumours began to regrow after a short delay following RT. However, these regrowing tumours showed signs of potential ulceration and these mice were

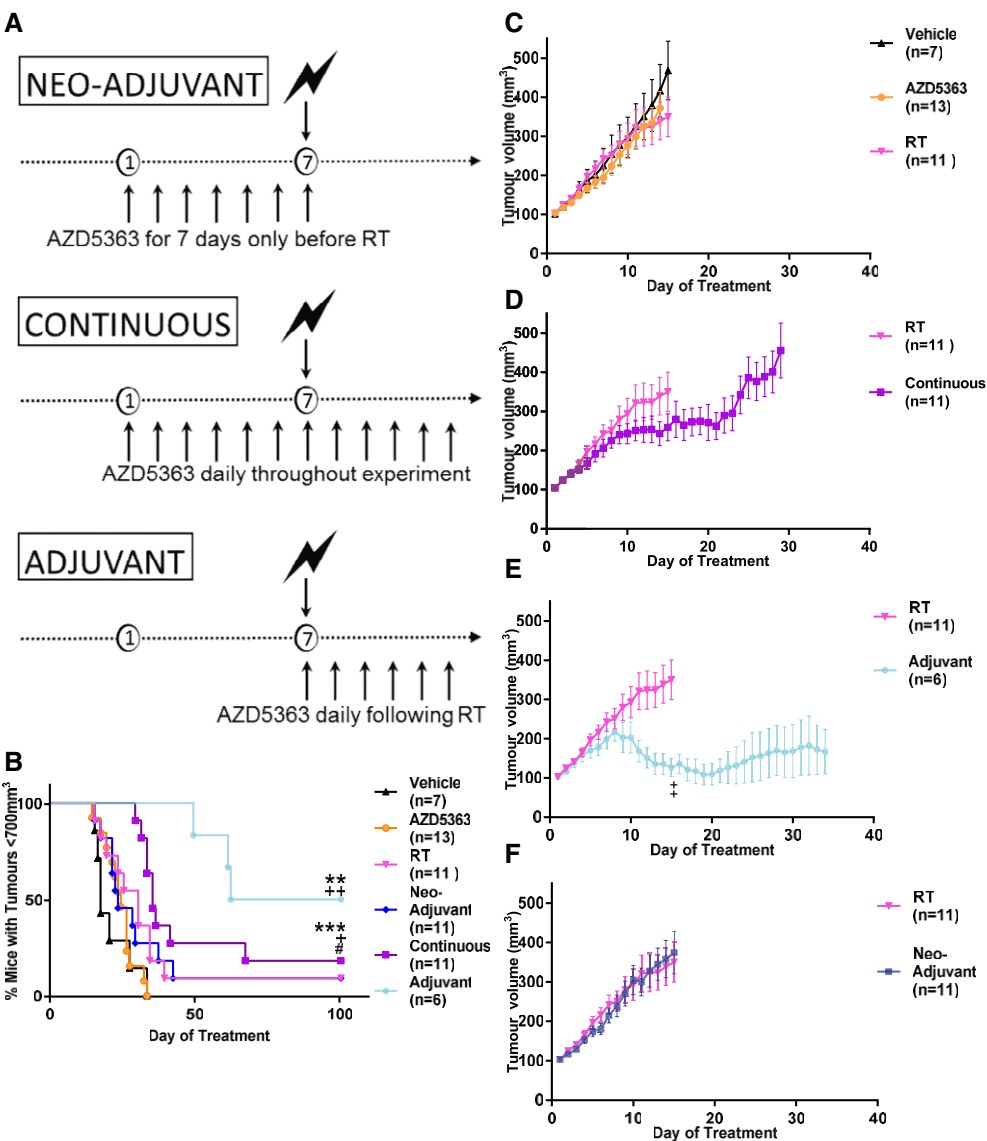

**Figure 2. Adjuvant AZD5363 following RT is more effective in improving tumour control in the FaDu head and neck cancer model than other treatment schedules.**

A   Diagram displaying the differing sequences of AZD5363 used in combination with RT in FaDu tumour-bearing mice.
B   Percentage of mice with tumour control on each experimental day. Plots show the combined data of two independent experiments (*n* = 6–13 mice/group).
C–F   Growth of FaDu tumours treated with AZD5363 (50 mg/kg BD) or 6 Gy RT alone (C), or in combination, with the drug given continuously (D), as an adjuvant (E) or as a neo-adjuvant (F). Plots show the combined data of two independent experiments (*n* = 6–13 mice/group).

Data information: In (B) data is presented as percentage mice with tumour control (tumour volume less than 700 mm³); *, significance compared to control mice; ⁺, significance compared to RT monotherapy; #, significance compared to adjuvant group: +/#*P* < 0.05, **/++*P* < 0.01, ***P* < 0.001. In (C–F) data are presented as mean tumour volumes ± SEM; ++*P* < 0.01. Statistical test in (B) is Gehan-Breslow-Wilcoxon test (*P* = 0.002), and in (E) is one-way ANOVA with Dunnett's *post hoc* test (*P* = 0.0001 and *P* = 0.038).
Source data are available online for this figure.

culled for welfare reasons. In no other treatment group (including vehicle and RT control groups) was this effect observed.

The effect of AZD5363 on the Akt pathway in FaDu xenografts was evaluated. The levels of pAkt, pGSK3β and pS6 in lysates made from tumours harvested 2 h after oral drug dosing were measured by Western blot (Appendix Fig S5A). Levels of pPRAS40 were also assessed in these samples by ELISA (Appendix Fig S5B).

AZD5363-induced hyperphosphorylation of Akt was demonstrated (Appendix Fig S5A), as is expected with an ATP-competitive inhibitor of Akt. However, some variability was observed in the effect on the downstream targets assessed. No reduction in pS6 was seen following treatment with AZD5363 (Appendix Fig S5A) whilst the level of pPRAS40 measurable in tumours from mice treated with AZD5363 was significantly reduced (*P* = 0.032; Appendix Fig S5B).

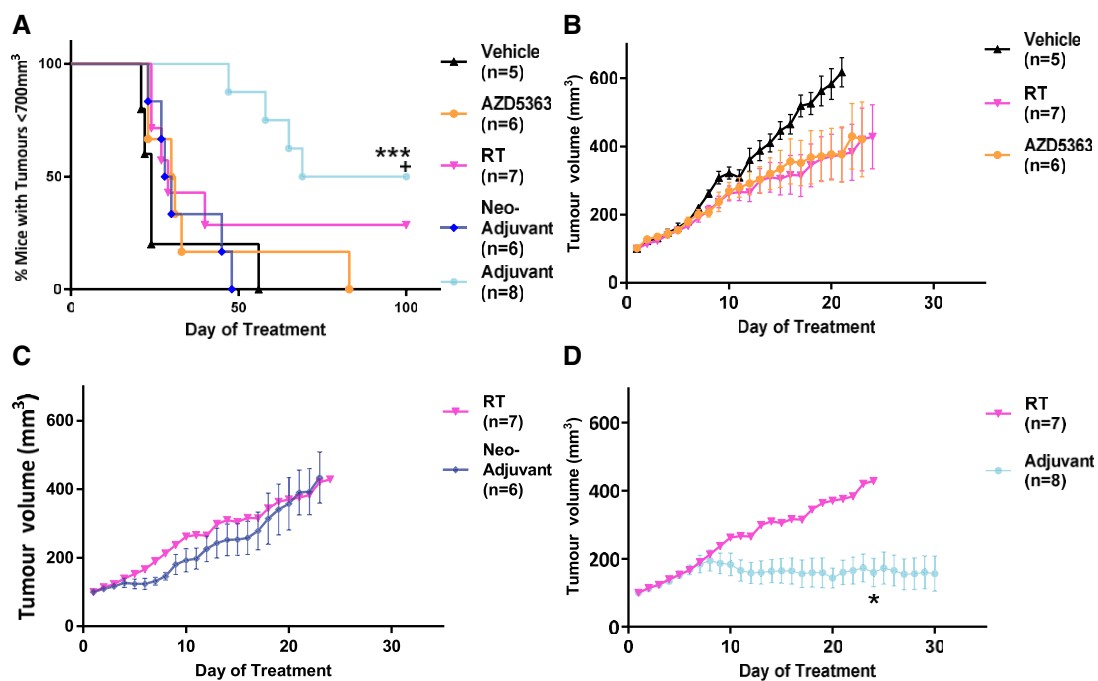

**Figure 3. Optimal tumour control is achieved in the PE/CA PJ34 model with adjuvant AZD5363 following RT.**

PE/CA PJ34 tumour-bearing mice were treated with 6 Gy RT either alone or in combination with AZD5363 (*n* = 6–13 mice/group).

A       Percentage of mice with tumour control on each experimental day.

B–D   Tumour volumes for single-agent AZD5363 and RT (B), neo-adjuvant (C) or adjuvant (D) AZD5363 (50 mg/kg BD).

Data information: In (A) data are presented as percentage mice with tumour control (tumour volume less than 700 mm³); *, significance when compared to control mice; +, significance when compared to RT monotherapy: +*P* < 0.05, ****P* < 0.001. In (C, D) data are presented as mean tumour volumes ± SEM; *P* < 0.05. Statistical test in (A) is Gehan-Breslow-Wilcoxon test (+*P* = 0.044 and ****P* = 0.009), and in (D) is one-way ANOVA with Dunnett's *post hoc* test (adjusted *P* = 0.043). Source data are available online for this figure.

## When given after RT, AZD5363 causes a reduction in tumour vessel density and an increase in tumour hypoxia

We hypothesised that the efficacy of AZD5363 in improving tumour control when given RT may result from effects on tumour vasculature. We initially carried out a limited exploratory study using FaDu tumours implanted into dorsal window chambers. Bright-field imaging was used every 2–3 days, and 14 days after implant, a functional vascular supply was seen to have been established in these tumours. The tumours were then given a single dose of 6 Gy RT, after which mice received twice-daily AZD5363 or vehicle. Imaging was continued every 2–3 days to allow measurement of total vessel length. Results suggested a marked reduction in total vessel length 7 days after RT in mice treated with adjuvant AZD5363 (Fig EV1A and B). Tumours which had received only RT outgrew the windows rapidly, within the first 3 days after treatment, necessitating early culling of that group and preventing comparison at the 7-day point between groups.

We next sought to further investigate tumour hypoxia and vascularity at that day 7 timepoint. Mice were inoculated with FaDu tumour cells, and once tumours reached 100 mm³, mice were treated with RT alone, or RT plus adjuvant or neo-adjuvant AZD5363 as in previous experiments (Fig 2A). Tumours were harvested 7 days after RT (the 14th day of treatment) and assessed

histologically by staining for CD31 and pimonidazole. Combination treatments were assessed against the RT alone control. Analysis of CD31 showed the vessel density of the adjuvant-treated tumours to be significantly reduced compared to the RT control (adjusted *P* = 0.0004; Fig 4A and B). There was no significant reduction found in the vessel density in the neo-adjuvant-treated group (Fig 4B). This observation was given further validation by assessment of hypoxia in these same tumours, through measurement of pimonidazole binding. Inverse to the findings with vascular density, a significantly higher percentage of pimonidazole-positive cells were found in the adjuvant AZD5363-treated mice compared to RT alone (adjusted *P* = 0.047; Fig 4C and D). Mice treated with the combination of neo-adjuvant AZD5363 and RT did not have a higher percentage of pimonidazole-positive viable tumour cells than occurred with RT alone (Fig 4D).

## AZD5363 acts to reduce HIF-1α and tumour-derived VEGF

In order to further examine the apparent anti-vascular effects of AZD5363 after RT, we sought to determine the effect of AZD5363 on tumour levels of VEGF and HIF-1α. Mice bearing FaDu tumours measuring 100 mm³ were commenced on 7 days twice-daily treatment with either AZD5363, or vehicle. On day 7 of treatment, 2 h after the 14th dose, all mice were culled and the tumours surgically

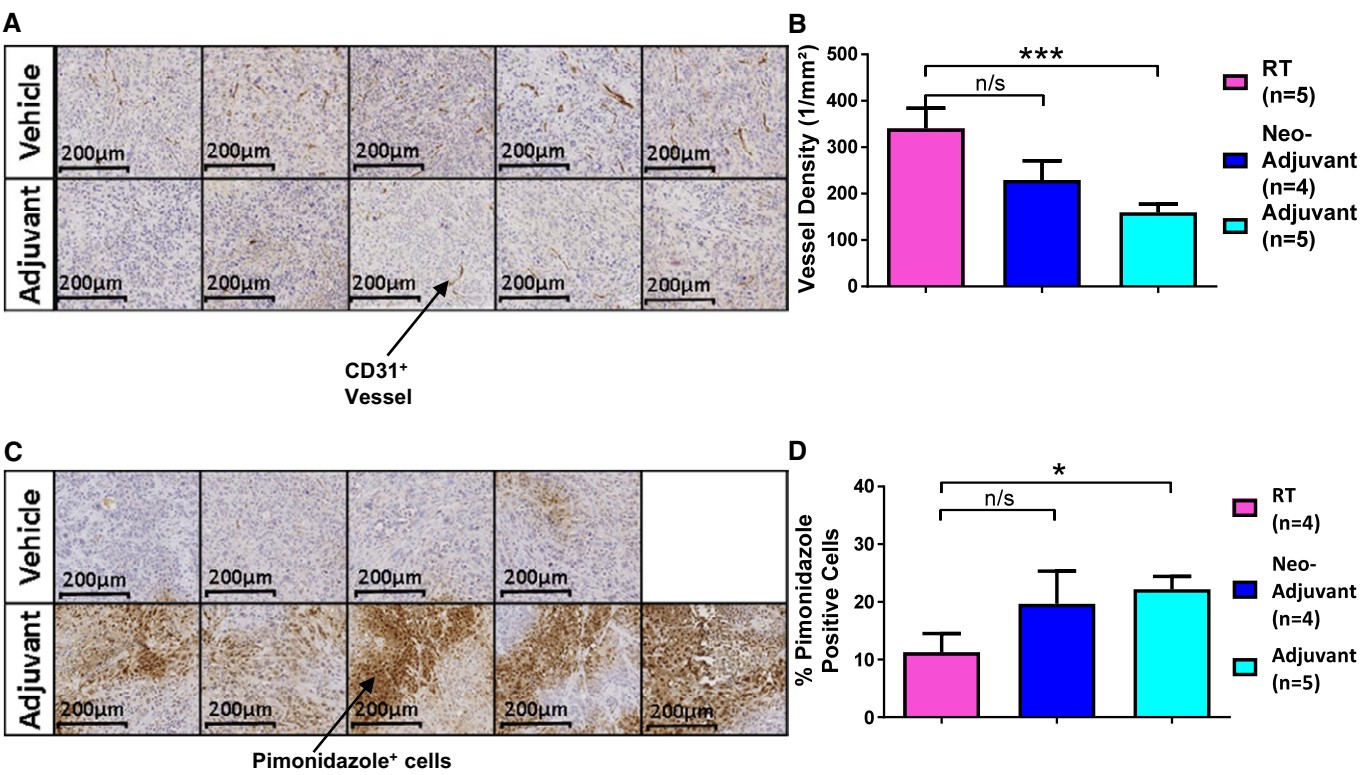

**Figure 4. When given after RT, AZD5363 causes a reduction in tumour vessel density, which is associated with an increase in tumour hypoxia.**

Histological analysis of FaDu tumours 7 days after treatment with 6 Gy RT alone, or in combination with neo-adjuvant or adjuvant AZD5363 (50 mg/kg BD). Tumour sections were stained with antibodies to detect CD31 or pimonidazole. Three sections were stained with each antibody, per tumour (*n* = 4–5 mice/group).

A   Sections demonstrating CD31 staining to allow visualisation of tumour vessels; 10× magnification.
B   Vessel density as measured in tumour sections.
C   Sections demonstrating pimonidazole staining to allow assessment of hypoxia; 10× magnification.
D   Percentage of pimonidazole-positive viable tumour cells.

Data information: (B, D) show mean levels ± SEM; *P < 0.05, ***P < 0.001. Statistical test in (B) and (D) is Kruskal–Wallis with Dunn's *post hoc* test (adjusted *P = 0.047, ***P = 0.0004).

excised for assessment. The effect of AZD5363 on HIF-1α was investigated by immunohistochemistry (IHC) in fixed tumour tissue. The percentage of HIF-1α$^+$ viable tumour cells was significantly reduced in the tumours from mice treated with AZD5363, relative to vehicle-treated mice (*P* = 0.004; Fig 5A and B). The levels of both human and mouse VEGF in lysates made from tumour tissue snap-frozen at the time of excision were then measured. AZD5363 treatment significantly reduced the levels of human VEGF in tumours compared with those found in tumours from vehicle-treated mice (846.7 ± 158.3 pg/ml vs. 1,435 ± 113.0 pg/ml, respectively; *P* = 0.01; Fig 5C). There was no detectable difference in the levels of murine-derived VEGF (Fig 5D).

### AZD5363 has no effect on tumour vascularity or hypoxia

Having established that AZD5363 has a direct effect on the level of pro-angiogenic proteins, we next investigated whether these changes resulted in functional alterations in tumour vascularity or hypoxia when AZD5363 was used as a single agent. Mice bearing FaDu tumours measuring 100 mm$^3$ were treated with twice-daily AZD5363, or vehicle. On the morning of day 6 of

treatment, the mice underwent FAZA PET/CT (18F-fluoroazomycin arabinoside, positron emission tomography, computerized tomography) imaging to provide an assessment of tumour hypoxia and perfusion. There was no significant difference in MTV between the treatment groups. There was no difference in initial FAZA distribution between groups, as might occur if significant differences in perfusion existed. In addition, there was no difference in tumour hypoxia as measured by tumour FAZA uptake (Fig 5E and F).

The day after the imaging, the mice were injected with pimonidazole, and 2 h after the 14$^{th}$ dose of vehicle or AZD5363, the mice were culled to allow histological confirmation of tumour hypoxia and vascularity. When tumour sections were stained with an anti-CD31 antibody to enable vessel visualisation, no difference in vessel density could be detected (Figs 5G and EV2A). When the percentage of pimonidazole-positive viable tumour cells was examined within these same tumour sections, no significant difference was found (Figs 5H and EV2B). This supports the FAZA PET/CT data in the conclusion that AZD5363, when used alone, does not alter tumour hypoxia or vascularity.

                                                    

**Figure 5. AZD5363 does not reduce vascularity or increase hypoxia when given as a single agent.**

FaDu tumour-bearing mice were treated with AZD5363 (50 mg/kg BD) or vehicle. On day 6 of treatment (after 11 doses of drug/vehicle), mice underwent tumour imaging with FAZA PET/CT. After 7 days of treatment (14 doses), tumours were assessed histologically for HIF-1α, CD31 and pimonidazole, and by ELISA for VEGF. Three sections were stained with each antibody per tumour (n = 4–5/group).

A   Representative tumour sections stained with anti-HIF-1α antibody (10× magnification).
B   HIF-1α levels relative to vehicle control mice.
C, D   ELISA measurement of human and mouse VEGF, respectively.
E   Example FAZA PET/CT of a FaDu tumour-bearing mouse.
F   18F-FAZA accumulation in tumours as measured by standardised uptake values (SUV) mean.
G   Tumour vessel density (CD31 staining).
H   Tumour hypoxia levels (pimonidazole staining).

Data information: Column charts give mean values ± SEM; n/s $P > 0.05$, *$P < 0.05$, **$P < 0.01$. In (B–H) statistical test is Mann–Whitney (*$P = 0.01$, **$P = 0.004$).

## Treatment with AZD5363 reduces proliferating vascular endothelial cells *in vivo*

We hypothesised AZD5363 might reduce the proportion of proliferating vascular endothelial cells. Such an effect could be expected to inhibit angiogenesis after RT, but potentially increase radioresistance if AZD5363 was given before RT. We therefore used the same tumours that had earlier been examined histologically for the effect of AZD5363 on tumour vessel density, pimonidazole binding and HIF-1α, to assess the effect of AZD5363 on vascular endothelial cell proliferation.

Sections were stained, firstly for the cytoplasmic marker CD31, to allow identification of both mature and immature vascular endothelial cells, and secondly, stained for the nuclear proliferative

marker, mouse Ki-67. Following this, DAPI was applied to aid nuclear identification (Fig 6A). The number of vascular endothelial cells co-expressing Ki-67 was then calculated as a percentage of the total number of vascular endothelial cells, with the aid of image analysis software. This analysis showed that the proportion of proliferating vascular endothelial cells was reduced in those tumours that had received treatment with AZD5363, compared to

those treated with vehicle ($P < 0.0001$; Fig 6B). There was no significant difference in the total number of CD31+ cells, as might have been anticipated from the lack of difference in vascular density observed between these tumour groups (Fig 5G). The total number of mouse Ki-67+ cells was significantly reduced throughout the stromal compartment ($P < 0.0001$; Fig 6D). Further IHC staining showed no difference in the percentage of human Ki-67+ tumour

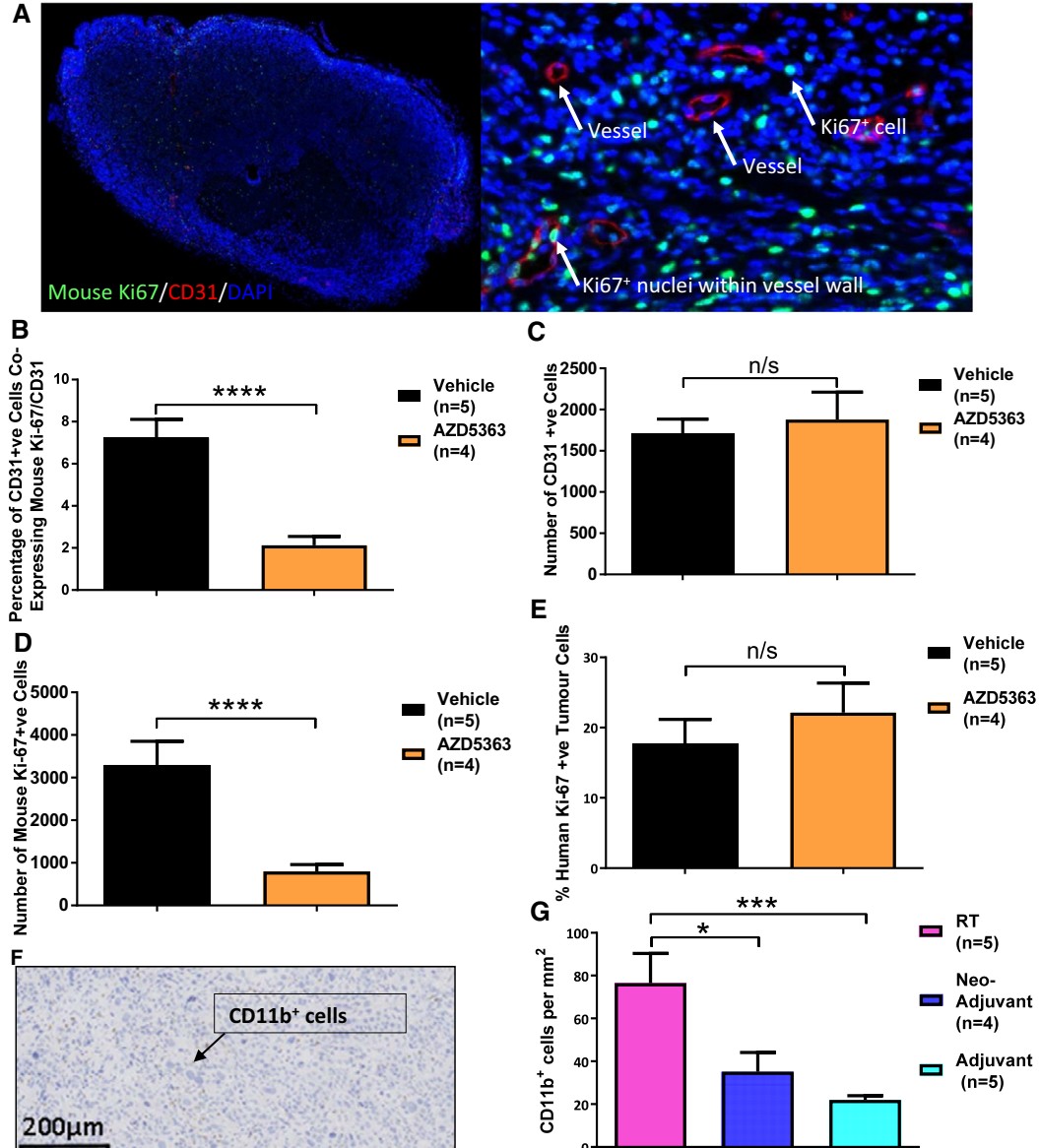

**Figure 6.  Treatment with AZD5363 reduces the proportion of proliferating vascular endothelial cells and following RT, the influx of CD11b+ bone marrow-derived cells.**

A–E  Tumours from FaDu xenograft-bearing mice treated with AZD5363 (50 mg/kg BD) or vehicle for 7 days were harvested for histological analysis and stained for co-expression of CD31 and mouse Ki-67 (A–D), or human Ki-67 (E); 3 consecutive sections stained and analysed; $n$ = 4–5/group. (A) Example of a whole (left) and magnified (right) section stained for CD31, mouse Ki-67 and DAPI (20× magnification). (B) Percentage of CD31+ cells co-expressing mouse Ki-67. (C) Number of CD31+ cells. (D) Number of mouse Ki-67+ cells. (E) Percentage of human Ki-67+ viable tumour cells.

F  Typical section of an RT control tumour stained with anti-CD11b antibody; 100× magnification.

G  Number of CD11b+ bone marrow-derived cells, per unit area of viable tumour; 3 consecutive sections stained and analysed; $n$ = 4–5/group.

Data information: In (B–E) and (G) data are presented as mean levels ± SEM; n/s $P > 0.05$, *$P < 0.05$, ***$P < 0.001$, ****$P < 0.0001$. Statistical test in (B–E) is Mann–Whitney test ($P = < 0.0001$) and in (G) is Kruskal-Wallis with Dunn's *post hoc* test (adjusted *$P = 0.025$, ***$P < 0.0004$).

cells with AZD5363 treatment (Figs 6E and EV2C). The absence of any difference is in keeping with the lack of effect of single-agent AZD5363 on tumour growth in this model (Fig 2C).

## Treatment with AZD5363 reduces the proliferation of human vascular endothelial cells *in vitro* but does not alter their radiosensitivity

We investigated whether the anti-proliferative effect of AZD5363 seen with mouse endothelial cells might similarly occur with the treatment of human vascular endothelial cells (HUVECs). Using BRDU uptake, we found a greater than 50% reduction in proliferation in HUVEC cells when treated with doses of between 2 and 3 μM AZD5363 for 96 h (Fig EV3A). Thus, AZD5363 appears to have a marked effect of the proportion of proliferating mouse and human vascular endothelial cells. However, when the BrdU assay was performed with and without the administration of 6 Gy RT, it was found that 1 μM AZD5363 did not provide a greater than additive reduction in the proliferation of HUVEC cells (Fig EV3B). This held true whether the AZD5363 was administered before, after or both before and after the dose of RT.

## Treatment with AZD5363 after RT reduces the intratumoural level of CD11b[+] myeloid cells

A HIF-mediated influx of bone marrow-derived cells (BMDCs) has previously been reported to be important in the recovery of tumour vasculature following RT, through a process termed vasculogenesis (Kioi *et al*, 2010). These cells are predominately CD11b[+] cells of myeloid origin. We therefore investigated whether AZD5363 might reduce this CD11b[+] population influx. Assessment of the number of CD11b[+] cells present within the viable tumour portion of the sections (Fig 6F and G) showed a significant reduction in the levels of these cells, in mice which had received adjuvant AZD5363 (adjusted $P = 0.0004$; Fig 6G) when compared to those that had received RT alone. The number of CD11b[+] cells present within tumours treated with neo-adjuvant AZD5363 was also significantly reduced, but to a lesser extent (adjusted $P = 0.025$; Fig 6G).

## Discussion

Here, we show that treatment with the potent Akt inhibitor AZD5363 permits long-term tumour control when given following RT, when neither drug nor RT alone has any effect on tumour growth. This greatly improved tumour control is demonstrated in two models of human cancer (FaDu and PE/CA PJ34) and is schedule dependent. Importantly, AZD5363 delivered as an adjuvant after RT led to markedly improved tumour control. Half of the mice that received this combination schedule had no measurable tumour mass on day 100 of treatment. In contrast, the administration of AZD5363 before RT failed to improve tumour control and continuous drug dosing (both before and after RT), led to only modest improvements in tumour control. AZD5363 has no effect on the intrinsic radiosensitivity of tumour cells, as demonstrated in a wide range of human cancer cells *in vitro*. Instead, AZD5363 appears to impact directly on the irradiated tumour microenvironment leading to markedly inhibited tumour vascularity and prevention of tumour regrowth.

*In vitro* experiments revealed that AZD5363 does not radiosensitise any of 15 different cell lines derived from tumour types commonly treated with RT in clinic. This was irrespective of the sensitivity of these cells to AZD5363 monotherapy, or the schedule of the drug relative to RT. Inhibitors of the Akt pathway have been shown to increase the radiosensitivity of human cancer cell lines (Martelli *et al*, 2003; Kim *et al*, 2005; Li *et al*, 2009; Chautard *et al*, 2010; Connolly *et al*, 2012). However, this effect does not occur with all compounds or in all cell lines (de la Pena *et al*, 2006; Narayan *et al*, 2014) and the timing and sequencing of drug and RT has also been suggested to be important in obtaining radiosensitisation (Kuger *et al*, 2013). The lack of an *in vitro* effect suggests the intrinsic radiosensitivity of tumour cells is not affected by Akt pathway inhibition with this compound; the effect *in vivo* must be explained by alteration of the tumour microenvironment.

Our results are in keeping with previous reports indicating that inhibitors of the PI3K/Akt pathway can have effects of tumour vascularity (Rubin *et al*, 1996; Gupta *et al*, 2005; Fokas *et al*, 2012; Kuger *et al*, 2013). Our preliminary dorsal window chamber experiment is simply hypothesis generating and suggests AZD5363 might lead to a reduction in tumour vascularity when administered for 7 days after RT. However, further work then supported the hypothesis that AZD5363 can alter tumour vascularity when administered after RT. In particular, as anticipated, we found that the effect of AZD5363 on pro-angiogenic tumour signalling revealed levels of HIF-1α were significantly reduced in drug-treated tumours, as were tumour-derived VEGF levels. In addition, it was interesting to note that as tumour cell Ki-67 levels were not reduced by AZD5363, the reduction in tumour-derived VEGF did not appear to be dependent on tumour cell senescence. RT-induced elevations of both VEGF HIF-1α are reported to confer radioprotection to tumour vasculature (Moeller *et al*, 2004; Harada *et al*, 2009). We observed that tumours from mice dosed with AZD5363 for 7 days after RT had significantly lower vessel densities and were significantly more hypoxic, than tumours from mice that received RT only.

We also found that the numbers of CD11b[+] cells present in adjuvant-treated mice were significantly lower than that found in mice treated with RT alone, when measured 7 days after RT. This is a potentially important observation given that CD11b[+] BMDCs are reported to be crucial in the recovery of tumours after RT, in part through the process of vasculogenesis (Russell & Brown, 2013). In addition, it has been reported that HIF-1α activity is necessary for the influx of BMDCs into the tumour microenvironment after RT (Kioi *et al*, 2010). Inhibition of HIF-1 after RT has been found to have profound effects on the growth of irradiated tumours (with constant shrinkage and no regrowth) whilst un-irradiated tumours are not affected (Kioi *et al*, 2010). These findings suggest that the effect of AZD5363 on this HIF-mediated immune response may be important in post-RT tumour control and may contribute to the speed and depth of response. This finding has not previously been described with a potent inhibitor of Akt, but might be anticipated by the known importance of Akt in the oxygen independent regulation of HIF-1 (Semenza, 2003). This work also provides rationale for future investigation as to the effects of AZD5363 on the wider immune contexture following RT in immunocompetent host tumour models.

Our analysis also revealed that AZD5363 reduced the proportion of proliferating tumour vascular endothelial cells *in vivo*, and HUVEC cells *in vitro*. This was despite an apparent lack of drug-alone effect

on the proliferation of tumour cells in the model investigated. These data provide further evidence for the effects of AZD5363 on tumour vasculature and previous reports that PI3K/Akt inhibitors can reduce the proliferation of HUVEC cells *in vitro* are consistent with this observation (Varma *et al*, 2005). We propose that when administered after RT, AZD5363 further abrogates vascular endothelial cell proliferation, already known to be inhibited by RT (Kozin *et al*, 2012). However, it is important to note that when used in combination with RT, AZD5363 did not have a greater than additive effect in reducing HUVEC proliferation *in vitro*, adding to the evidence that it is microenvironmental, rather than cellular, changes that bring about the improvement in tumour control with AZD5363.

The anti-proliferative effect of AZD5363 on vascular endothelial cells may also help to explain the observation that AZD5363 failed to improve tumour control when given prior to RT (Hall & Giaccia, 2012). Our results suggest that the relative radioresistance induced by the use of AZD5363 prior to RT might be explained at least in part by anti-proliferative effects on tumour vasculature endothelial cells. It is generally accepted that the high proliferative rate of tumour vascular endothelial cells contributes to the radiosensitivity of these vessels, relative to those occurring in normal tissues (Barker *et al*, 2015). Therefore, it is reasonable to conclude that inhibition of endothelial cell proliferation may be undesirable prior to RT.

The optimal AZD5363/RT combination schedule is different to that proposed with other inhibitors of the PI3K/Akt pathway, in which continuous treatment (before and after RT) is reported to be required to allow a process termed "vascular normalisation" in which hypoxia is reduced (Fokas *et al*, 2012; Hill *et al*, 2016). However, there are reports of other drugs impacting on tumour vasculature in combination with RT (such as inhibitors of HIF-1 and VEGF) that have demonstrated increased anti-tumour efficacy when administered after RT (Zips *et al*, 2003; Williams *et al*, 2004; Harada *et al*, 2009). Furthermore, it is known that vascular responses to PI3K inhibitors in preclinical models have diversity in the impacts on vessel density and tumour oxygenation (Soler *et al*, 2015). It is interesting to note that adjuvant drug administration with other inhibitors of BMDC influx has been shown to be successful in improving tumour control with a similar depth and duration of response (Kioi *et al*, 2010).

When compared to the observations after RT, it was interesting to note that single-agent AZD5363 had no appreciable effect on tumour vascular density or hypoxia (as measured by FAZA PET/CT or pimonidazole binding). This is in contrast to some other reports with PI3K/Akt pathway inhibitors (Rubin *et al*, 1996; Gupta *et al*, 2005; Fokas *et al*, 2012; Kuger *et al*, 2013). Our data therefore highlight the importance of considering scheduling when using inhibitors of this pathway alongside RT and the need to question assumptions from data generated with other compounds seemingly targeting similar signalling pathways.

The phase I trial of AZD5363 demonstrated an acceptable safety and tolerability profile (Banerji *et al*, 2013). The main side effects of the compound include diarrhoea and hyperglycaemia, which were found to be dose related. The equivalent preclinical exposure of the recommended phase II dose for AZD5363 for continuous dosing (320 mg BD) exceeds the preclinical 50 mg/kg BD exposure used in this work, which suggests the required dosing for use in the combination with RT will be safe and tolerable. The dose of RT used in this work was also lower than that needed for effective tumour cell kill

*in vivo* indicating that an improved therapeutic ratio can be achieved. In addition, given that the optimal sequence for combining AZD5363 and RT appears to be as adjuvant after RT, the risk of AZD5363 unexpectedly increasing RT-associated side effects will be minimised.

This work has provided proof of principle that AZD5363 can result in long-term tumour control when given after RT in the treatment of two models of head and neck cancer. The work will form the basis for translating the concept of combining AZD5363 with RT into clinical evaluation which will be aided by generating preclinical data demonstrating enhanced therapeutic effects with fractionated RT. We believe this will be effective since AZD5363 produces oxygen independent reduction in HIF-1α, so that crucial signalling in hypoxic and re-oxygenating areas will be similarly affected.

In the clinical evaluation of the AZD5363/RT combination, it will also be important to investigate potential biomarkers. Demonstration of pPRAS40 inhibition in the hair follicle was found to be a useful pharmacodynamic biomarker in the phase I clinical trial (Banerji *et al*, 2013). In addition, PI3K pathway oncogenic mutations have been shown to predict tumour cell sensitivity to AZD5363 monotherapy *in vitro* and *in vivo* (Davies *et al*, 2012; Banerji *et al*, 2013). Whilst hair follicle examination may prove a useful priori biomarker, our work has demonstrated that monotherapy drug effect on tumour growth was not required to permit combination efficacy with AZD5363 and RT. Valuable biomarkers of the combination effect and predictors of response could include markers of vascular endothelial cell proliferation and high HIF-mediated gene expression in the pretreatment biopsy samples. In addition, intratherapy-imaging of vascular function using dynamic contrast-enhanced MRI (DCE-MRI) would also be a potential biomarker in certain patient tumour groups.

In summary, this study demonstrates that AZD5363 can produce significant and durable improvements in tumour control when used as an adjuvant treatment after RT. Our mechanistic investigations have revealed that the efficacy of AZD5363 in combination with RT does not depend on an alteration of cellular radiosensitivity, but instead on the alteration of the tumour microenvironment. AZD5363 reduces key pro-angiogenic and vascular radioprotective proteins (VEGF and HIF-1α), inhibits vascular endothelial cell proliferation and reduces the recruitment of CD11b[+] BMDCs. These effects contribute to a marked reduction in tumour vascularity when AZD5363 is administered post-RT and a failure of tumour regrowth. The importance of the correct sequencing of AZD5363 relative to RT has been clearly demonstrated, which has implications for the use of Akt pathway inhibitors in future studies. As seen here, the duration and improvement in tumour control demonstrated when AZD5363 was administered after RT is encouraging and provides optimism in translating these finding to clinical trials of RT and AZD5363 combinations.

## Materials and Methods

### Cell culture

Details of the original source and culture conditions of all cell lines used are contained in Appendix Table S1. Reagents were obtained from Invitrogen. For use in this research, cells were obtained directly from a cell bank that performs cell line characterisations by

short tandem repeat analysis and were passaged for fewer than 6 months after resuscitation. Cells were routinely tested for mycoplasma in-house using PCR detection.

### AZD5363 formulation

AZD5363    [(S)-4-amino-N-[1-(4-chlorophenyl)-3-hydroxypropyl]-1-(7H-pyrrolo[2,3-d]pyrimidin-4-yl)piperidine-4-carboxamide]    was prepared as a 10 mmol/l stock solution in DMSO (dimethyl sulphoxide). The final concentration of DMSO was < 0.1% in all *in vitro* assays. For administration *in vivo*, AZD5363 was solubilised in a 10% DMSO 25% w/v Kleptose HPB (Roquette) buffer, and given by oral gavage.

### Western blot analysis

AZD5363-treated tumour cells were lysed on ice with a complete cell extraction buffer, made by the addition of cOmplete™ EDTA-free protease inhibitor cocktail (Sigma-Aldrich) and PMSF (phenylmethanesulphonyl fluoride) to cell extraction buffer (Invitrogen). Snap-frozen tumour portions were homogenised using a Fastprep™ homogeniser (MP Biomedicals) in the same complete cell extraction buffer. Western blot analysis then proceeded as already published (Davies *et al*, 2012). Antibodies are detailed in Appendix Table S2.

### Enzyme-linked immunosorbent assay (ELISA)

Three different commercially purchased ELISA kits were used during the course of this research to measure proline-rich Akt substrate 40 (pPRAS40) and VEGF (Appendix Table S3), all according to manufacturers' instructions.

### Viability assay

Tumour cells were treated with 0.1–10 μM AZD5363 for 96 h following which an MTT (3-[4,5-dimethylthiazol-2-yl]-2,5 diphenyltetrazolium bromide) assay was performed as previously published (Lunt *et al*, 2010). Dose-response curves were drawn according to the equation:

$$Y = \text{Bottom} + \frac{(\text{Top} - \text{Bottom})}{(1 + 10^{X - \text{LogIC50}})}$$

### Clonogenic survival assays

Clonogenic assays were conducted in all tumour cell lines with 1 or 10 μM AZD5363, as previously published (Senra *et al*, 2011).

### Flow cytometric analysis of cell cycle with propidium iodide (PI)

FaDu cells (200,000 cells/flask) were seeded in T-25 flasks. The following day the flasks were treated with 1 μM AZD5363, and 2 h later, 4 Gy RT was delivered to the cells at a rate of 0.952 Gy/min (Faxitron). After 24 h, the contents of the flasks media were removed, centrifuged (400 *g* for 5 min), and the cell pellet was resuspended in 300 μl hypotonic PI solution (50 μg/ml PI (Sigma-Aldrich), 0.1% (w/v) sodium citrate, 0.1% (v/v) Triton X-100) and left overnight at 4°C. 10,000 events were then analysed using a

FACSCalibur cytometer (BD Biosciences), and the data examined using the software Summit version 4.3 (Dako).

### Initiation of FaDu and PE/CA PJ34 tumours

Animal study protocols were approved by the Institutional Ethics Committee and the Home Office (project licenses 40/3212 and 70/7760) and designed in accordance with the Scientific Procedures Act (1986) and the 2010 guidelines for the welfare and use of animals in cancer research (Workman *et al*, 2010). Animals were housed in the University of Manchester's Biological Services Facility. Mouse numbers were determined by the calculation of group sizes needed to obtain a biologically relevant 40% change from control (assuming the significance level is set at 5%) with a power of > 80%, based on previous growth data using InVivostat software (www.invivostat.co.uk).

Tumour cells in exponential phase were prepared to the desired concentration (FaDu, $2 \times 10^6$ cells in 0.1 ml serum-free media; PE/CA PJ34, $1 \times 10^7$ cells in 0.1 ml PBS). To initiate tumour xenografts, 0.1 ml of cell suspension was implanted subcutaneously on the back of 8–12 weeks Female CD-1® Nude Mice (Charles River) weighing 22–28 g. Mouse weights were monitored daily, and once palpable tumours were formed, the volumes were measured daily by the formula: tumour width × length × depth.

### *In vivo* treatment schedules

Mice bearing tumours measuring 100 mm³ in size were randomised into AZD5363 (50 mg/kg, PO, BD) alone, RT alone, vehicle alone and combination treatment groups. Mice receiving neo-adjuvant treatment were dosed with AZD5363 for 7 days before RT, receiving the last dose of AZD5363 2 h before RT. The mice treated according to a continuous dosing schedule received AZD5363 for 7 days before, and continuously after, RT. The mice in the adjuvant group started treatment with AZD5363 from 2 h after RT and were then dosed twice daily until the end of the experiment. Mice were culled when the tumour volume exceeded 700 mm³, or if this end-point was not reached, on day 100 after the beginning of treatment. Tumour control was considered to be lost if the tumour volume exceeded 700 mm³, and long-term tumour control considered achieved if animals had no measurable tumour mass on day 100 after the beginning of treatment. MXR-320/36 X-ray apparatus (320 kV, Comet AG) was used to deliver 6 Gy RT (2 Gy/min) to non-anaesthetised, tumour-bearing mice. The animals were restrained in jigs which used lead shielding to allow local RT to be delivered to the tumour, whilst protecting other tissues. Regular calibration was provided by the Medical Physics Department of the Christie Hospital (Manchester, UK). Where administered, pimonidazole (Hypoxyprobe, Burlington, USA) was given by intraperitoneal (i/p) injection 2 h before culling in all experiments.

### Dorsal window chamber model

Dorsal window chambers were set up as previously described (Williams *et al*, 2007). Treatments were initiated when substantial vascularisation was visualised within the tumour by bright-field microscopy (~14 days after implantation of $5 \times 10^7$ FaDu cells). Mice received 6 Gy RT, delivered to the tumour, before

commencing on 50 mg/kg AZD5363 or vehicle BD 2 h later. Repeat imaging was carried out on days 2, 4 and 7 after RT. Repeat imaging on vehicle/RT-treated tumours was not possible beyond day 2 due to rapid outgrowth of the tumour window. Images at a 40× magnification were then inverted and the contrast enhanced using Photoshop Express (Adobe) and then analysed using AngioTool software (Zudaire *et al*, 2011).

### FAZA PET/CT imaging

Mice treated with either AZD5363 50 mg/kg, PO, BD or vehicle for 6 days underwent FAZA PET/CT. A CT scan was first performed following which 25 MBq of 18F-FAZA was injected i/v. A 30-min dynamic PET scan was performed immediately thereafter. Four hours post-injection, a 20-min static PET scan was conducted. Mean standardised uptake values (SUVmean = (activity/ml tissue)/(injected activity/body weight), ml/g) were calculated for each tumour.

### Immunohistochemistry

After formalin fixation, xenograft tumour samples were paraffin-embedded and freshly sectioned before staining. Sections were stained for CD31, pimonidazole, HIF-1$\alpha$, human Ki-67 and CD11b (Appendix Table S4), before counterstain with haematoxylin. Stained sections were analysed using Tissue Studio image analysis software (Definiens) in the Advanced Imaging Facility of the CRUK Manchester Institute (UK).

### Immunofluorescence

Fluorescent staining of formalin-fixed paraffin-embedded tumour sections for CD31 and mouse Ki-67 (Appendix Table S4) was carried out on a Discovery automated slide staining system (Ventana). A TSA® Plus multi-fluorophore (Cyanine 3 and Fluorescein) detection system kit (PerkinElmer) was used to amplify and visualise the multiple signals, as per manufacturer's instructions. After staining, samples were mounted using ProLong® Gold Antifade Mountant with DAPI (4′,6-diamidino-2-phenylindole) (Life Technologies). Stained sections were analysed using Tissue Studio.

### Human vascular endothelial cell (HUVEC) proliferation assay

HUVEC cells (1,000 cells/well) were seeded in a gelatin-coated 96-well plate and treated with doses of AZD5363 ranging from 0.1 to 10 μM for either 48 or 96 h. A colorimetric BrdU (bromodeoxyuridine) cell proliferation ELISA kit (Roche) was then used as per manufacturer's instructions. In a second experiment, HUVEC cells were plated then treated with AZD5363 for 2 h before, 2 h before and 96 h after, or for 96 h after a single 6 Gy dose of RT. The BrdU assay was then performed.

### Statistical analysis

The effect of AZD5363 on PRAS40 phosphorylation, and of differing sequences of AZD5363 on the surviving fraction of FaDu cells was analysed using one-way ANOVA with Dunnett's test to adjust for multiple comparisons. The Gehan–Breslow–Wilcoxon test was used

### The paper explained

#### Problem

Radiotherapy is an important cancer treatment, delivered to approximately 50% of cancer patients. However, relapses following potentially curative radiotherapy treatment are common, causing significant mortality and morbidity. In an effort to improve clinical outcomes, there has been a growing interest in combining radiotherapy with drugs that target the molecular signalling pathways thought to be important in the response of tumours to radiotherapy. One such target is Akt, and in this study, we have looked at the effect of combining the potent Akt inhibitor AZD5363 with radiotherapy in mouse models of cancer.

#### Results

Our recent study has demonstrated that daily AZD5363 improves long-term tumour control when commenced 2 h after a single dose of radiotherapy in two tumour models of head and neck cancer (FaDu and PE/CA PJ34). Interestingly, we also demonstrated that administration of AZD5363 up until 2 h before radiotherapy failed to improve tumour control and continuous drug dosing (both before and after radiotherapy) leads to only modest improvement. Whilst both models expressed pAkt, *in vitro* experimentation revealed that AZD5363 has no effect on the intrinsic radiosensitivity of tumour cells. This was irrespective of the dose of AZD5363, the extent of drug-alone effect on tumour cell viability, or the sequence in which drug and radiation were delivered. Further *in vivo* investigation found the positive effect of AZD5363 on radiotherapy response is due to an impact on the irradiated tumour microenvironment. Whilst AZD5363 alone does not alter tumour vascularity or oxygenation as measured by histological and FAZA PET-CT methods, it has dramatic effects on the response of tumour vasculature to radiotherapy. When given after radiotherapy, AZD5363 reduces tumour vascular density, the influx of pro-vasculogenic myeloid cells and tumour regrowth. AZD5363 also inhibits vascular endothelial cell proliferation *in vitro* and *in vivo*, which may contribute to the schedule dependency of the tumour control effect.

#### Impact

This study has demonstrated the potential of Akt inhibition to significantly improve long-term tumour control following radiotherapy. The experiments performed in this work have also demonstrated the importance of establishing the correct sequence of radiotherapy and drug when using AZD5363 in patients receiving radiotherapy. AZD5363 is a promising clinical candidate providing the exciting possibility of early translation of this work into clinical trial.

to analyse the tumour control experiments *in vivo*, where time to a tumour volume of 700 mm$^3$ reached was evaluated. Group MTVs were plotted graphically until the last day of possible comparison (an animal was culled from the group) and analysed on this day using unpaired *t*-tests or one-way ANOVA with Dunnett's test to adjust for multiple comparisons. Tumour volumes were log-transformed to stabilise variance and remove size dependency; variance was unequal between groups as it increases with tumour volume. The Mann–Whitney test was used to assess the effects of AZD5363 on tumour levels of all markers, unless adjustment for multiplicity was required, in which case the Kruskal–Wallis test was used, followed by Dunn's multiple comparisons test. The effect of AZD5363 on the surviving fraction of cervical cell lines was tested using unpaired *t*-tests. In order to analyse the effect of AZD5363 on cell cycle, with and without radiation, the Kruskal-Wallis test was used, followed by Dunn's multiple comparisons test. All statistical analyses were performed using GraphPad Prism 6 software (GraphPad Software, Inc).

**Expanded View** for this article is available online.

## Acknowledgements

The authors thank the members of the Cancer Research UK Manchester Institute Histology and Advanced Imaging Core Facilities, and also the members of the University of Manchester Biological Support Facility for their help with this project. In addition, we wish to thank Alison Smigova for her assistance with animal imaging, and Phillippa Dudley and Neil James for their help with sample analysis. We are grateful to the Manchester Cancer Research Centre for funding this work via grants from AstraZeneca and Cancer Research UK (D1330/C00032).

## Author contributions

EJS, KJW, IJS and TMI designed the study and the experiments and wrote the manuscript. EJS and BAT performed the majority of the experiments with the help of DM who carried out *in vitro* studies with HUVEC cells. DMF performed the FAZA PET-CT imaging and analysis. BRD and DM provided scientific expertise helpful to the design of experiments. All authors read and commented on the manuscript.

## Conflict of interest

T.M. Illidge and I.J. Stratford received research grant support from AstraZeneca. B.R. Davies is an employee and shareholder of AstraZeneca.

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
