## [Review Process File · EMBO Molecular Medicine]

Akt inhibition improves long-term tumour control following radiotherapy by altering the microenvironment

Emma J. Searle, Brian A. Telfer, Debayan Mukherjee, Duncan M. Forster, Barry R. Davies, Kaye J. Williams, Ian J. Stratford & Tim M. Illidge

Corresponding author: Tim M. Illidge, University of Manchester

Review timeline:

Submission date:	16 March 2017
Editorial Decision:	12 May 2017
Revision received:	04 September 2017
Editorial Decision:	20 September 2017
Revision received:	27 September 2017
Accepted:	28 September 2017

Editor: Roberto Buccione

Transaction Report:

1st Editorial Decision

12 May 2017

Thank you for the submission of your manuscript to EMBO Molecular Medicine. We have now heard back from two Reviewers whom we asked to evaluate your manuscript.

We are very sorry that it has taken such an unusually long time to provide you with a decision on your manuscript. In fact, we experienced significant difficulties in securing willing and appropriate reviewers and in obtaining their evaluations in a timely manner. Also, I was unable to obtain a third evaluation and therefore I am now proceeding based on the two available, consistent evaluations. I trust that the inevitable frustration due to this delay will be somewhat tempered by the fact that the Reviewers are quite supportive and, in my opinion, offer valuable suggestions to improve the impact, strength and translational relevance of your manuscript.

I will not go into much detail, as the comments are clear in my opinion and should not prove too challenging to address.

Reviewer 1 would like you to explain and/or further elaborate on a number of topical points and would like you to address further the potential toxicity of the RT-drug combination. Reviewer 2 also suggests a number of data presentation improvements and clarifications, and would like you to experimentally show that circulating endothelial progenitor cells in the animal are sensitive to the ADZ5363, IR or the combination.

In conclusion, while publication of the paper cannot be considered at this stage, we would be pleased to consider a revised submission, with the understanding that the Reviewers' concerns must be addressed with additional experimental data where appropriate and that acceptance of the manuscript will entail a second round of review.

Please note that it is EMBO Molecular Medicine policy to allow a single round of revision only and that, therefore, acceptance or rejection of the manuscript will depend on the completeness of your

responses included in the next, final version of the manuscript.

As you know, EMBO Molecular Medicine has a "scooping protection" policy, whereby similar findings that are published by others during review or revision are not a criterion for rejection.

Please note that EMBO Molecular Medicine now requires a complete author checklist (<http://embomolmed.embopress.org/authorguide#editorial3>) to be submitted with all revised manuscripts. Provision of the author checklist is mandatory at revision stage; It is designed to enhance and standardize reporting of key information in research papers and to support reanalysis and repetition of experiments by the community. The list covers key information for figure panels and captions and focuses on statistics, the reporting of reagents, animal models and human subject-derived data, as well as guidance to optimize data accessibility. The Author checklist will be published alongside the paper, in case of acceptance, within the transparent review process file.

Last, but not least, please carefully conform to our author guidelines (<http://embomolmed.embopress.org/authorguide>) to ensure rapid pre-acceptance processing in case of a favorable outcome on your revision.

I look forward to seeing a revised form of your manuscript in due time.

***** Reviewer's comments *****

Referee #1 (Remarks):

This is an excellent piece of work by a strong radiobiology laboratory and addresses a unique aspect of RT-DRUG combinations when targeting AKT signaling: a pathway associated with poor local control in HEENT tumours.

The authors show convincingly that the AKT-inhibitor AZD5363 (AKTi) was much more effective as an adjuvant treatment than when given as a neoadjuvant to RT in two HEENT models (FaDu and PE/CA PJ34); importantly-in some animals-this led to complete local control. This effect was independent of the effects on the intrinsic radiosensitivity of cell lines *in vitro*. The adjuvant effect was associated with altered density, tumour vasculature endothelial function and correlated with an influx of CD11b+ myeloid cells *in vivo*.

Overall, the study is well conducted in two independent HEENT models with similar findings.

I have a few comments and suggestions that would make the publication's conclusions stronger:

- 1) Are there effects of AKT signaling on the cell cycle phase at the time of irradiation - could the sensitization *in vivo* also be due to altered cell cycle phase after prolonged *in vivo* treatments ? Have the authors completed *ex vivo* clonogenic survival curves with 6 or 10Gy to show radiosensitivity *in vivo* is not affected ?
- 2) Can the authors speculate or have data on the results if fractionated radiotherapy was used and the effects of increasing hypoxia on these intra-tumoural sensitivity with the agent ?
- 3) What biomarkers should be used for the PD approaches to the use of this agent in clinical trials ? Which patients might or might not benefit and how would one test *a priori* or *intra-therapy* for its continued adjuvant use ? This is important to show the translational aspects of the work.
- 4) What toxicity -if any- did the RT-drug combo elicit, or could elicit, with results of this study using the combined modality approach ? Please comment on the therapeutic ratio and data pertaining to it if available.

Referee #2 (Remarks):

Searle et. al. provide an interesting study that systematically evaluates how blockade of Akt with

ADZ5363 and treatment with ionizing radiation is not effective on tumor cells but alters the function of the cells in the tumor microenvironment.

There are some points of clarification and interpretations that need to be address to support the claims.

1. For the analysis of endothelial cells with ADZ5363 and IR the authors should consider constructing an isobologram to assess if there is an additive or synergistic effect.
2. While the data on ADZ5363 on tumor cells shows that it inhibits Akt in tumor cells alone, it is unclear if this compound with IR is still effective as no western blots were provided.
3. Is it possible that a subpopulation of tumor cells with IR+ADZ5363 undergo senescence, which would explain the decrease in VEGF?
4. The Figure S1 legend does not reflect the text on page 13 top paragraph.
5. It was not clear in Figure S5A and B if there was IR treatment alone. It was not clearly marked in the Figure.
6. In Figure 6 the data shows CD31 staining for vessels but it is unclear if the analysis was to measure all vessels. IT might be important to distinguish between mature vessels versus immature vessels.
7. The author suggesting that endothelial cells are most sensitive to ADZ5363 and IR. Could the author substantiate these finding by providing evidence that circulating endothelial progenitor cells in the animal are sensitive to the ADZ5363, IR or ADZ5363 and IR. This would provide evidence that source of the endothelial cells are from the bone marrow versus the mature vasculature.

1st Revision - authors' response

04 September 2017

Many thanks for forwarding the helpful and insightful comments from the reviewers. We are grateful for the opportunity to revise and enhance the manuscript in light of this feedback and suggestions. We have done our best to address the specific comments made by the reviewers alongside the changes made and these are outlined point by point below.

Referee #1:

1) Are there effects of AKT signaling on the cell cycle phase at the time of irradiation - could the sensitization in vivo also be due to altered cell cycle phase after prolonged in vivo treatments? Have the authors completed ex vivo clonogenic survival curves with 6 or 10Gy to show radiosensitivity in vivo is not affected?

This is a potentially important point and we agree that Akt inhibition with AZD5363 could have an effect on cell cycle at the time of irradiation. We have investigated this in FaDu cells and did not originally include our data as we found no evidence of an effect of AZD5363 on cell cycle measured by flow cytometric analysis with propidium iodide. In the light of the reviewers questions this data has been added into **Appendix figure S3J and in the manuscript (page 5, para 4)**. This data demonstrates that non-toxic concentrations of AZD5363 have no effect on cell cycle, either alone, or 24hrs after 4 Gy irradiation. This is in keeping with our hypothesis that the impact of AZD5363 on the response of tumours to radiotherapy does not result from a direct effect of cellular radiosensitivity, but is due to an impact on the tumour microenvironment. These results are consistent with published results on cell cycle analysis of prostate cancer cell lines where in PC3/DU145 cells, a 10 μ M concentration of AZD5363 had no cytotoxic effect and this was accompanied by no effects on cell cycle (Lamoureux et al, 2013). The MTT assays we performed revealed FaDu to be an insensitive cell line, and as such our results are in keeping with this publication. Our exhaustive *in vitro* experiments clearly demonstrated no impact on intrinsic radiosensitivity. This, when coupled with a lack of cell cycle effect, meant that while we did consider the possibility of assessing *ex vivo* clonogenic survival early on in our program, we rationalised to focus on the microenvironment.

2) Can the authors speculate or have data on the results if fractionated radiotherapy was used and the effects of increasing hypoxia on these intra-tumoural sensitivity with the agent?

Whether AZD5363 will improve long-term tumour control after fractionated radiotherapy is indeed an important question and will be the subject of future planned work. However, this work is beyond the scope of this proof of principle initial study. We would speculate that as the mechanism by which AZD5363 improves control after radiotherapy is at least, in part, derived from the effects of

AZD5363 of the oxygen independent regulation of HIF, that we are optimistic of an effect following fractionated radiotherapy. We have addressed this with **additional comments in the discussion (page 17, para 2)**.

3) What biomarkers should be used for the PD approaches to the use of this agent in clinical trials? Which patients might or might not benefit and how would one test a priori or intra-therapy for its continued adjuvant use? This is important to show the translational aspects of the work.

Developing suitable biomarkers is also an important part of our future work and in informing the translation of this work into early phase clinical trials. In the phase I trial investigating AZD5363 in advanced solid tumours pPRAS40 inhibition in the hair follicle was successfully used to demonstrate targeting of the Akt pathway in patients given AZD5363 (Banerji et al, 2013). In addition, Akt1 and PIK3CA gene mutations were found to be associated with the largest responses to AZD5363 in terms of tumour reduction. However, both of these approaches might be expected to only inform to a limited degree which patients might benefit from a combination approach of AZD5363 with radiotherapy. As both mouse models were relatively insensitive to AZD5363 monotherapy, it appears that monotherapy efficacy is not required for the combination of AZD5363 and radiotherapy to produce successful long term tumour control. Whilst on-target Akt inhibition via examination of the hair follicle may be an *a priori* test, other potential biomarkers will need investigation and validation in relation to combination efficacy. We consider that potential biomarkers may include vascular endothelial cell proliferation and high HIF-mediated gene expression in the pre-treatment biopsy sample. Intra-therapy-imaging of vascular function (DCE-MRI) would also be a potential biomarker for future investigation. **We have added a comment on these approaches into our discussion (page 17, para 3)**.

4) What toxicity -if any- did the RT-drug combo elicit, or could elicit, with results of this study using the combined modality approach? Please comment on the therapeutic ratio and data pertaining to it if available.

There were no discernible differences in toxicity between the combination, single treatment and control arms in either experiment. Mouse weight data has now been included in supplementary figure S4A. The adverse effects seen in the Phase I trial with AZD5363 included hyperglycaemia and diarrhoea and appeared dose related. The recommended phase II dose for AZD5363 is 320 mg BD for continuous dosing. The dosing of 400 mg BD is approximately equivalent to 100 mg/kg BD preclinical exposure (in the mouse), indicating equivalent drug doses to the 50mg/kg BD pre-clinical exposure used in our work are achievable. The radiation dose used in our work was not high enough to cause any effect on tumour growth alone, therefore in our studies the benefit of AZD5363 is achieved at no "cost" with respect to normal tissue toxicity. Although this infers high therapeutic ratio, this is a very large claim to make from a xenograft model in a short term experiment. Given our proposal that AZD5363 should be given as an adjuvant to RT thereby negating potential for exacerbating effects during RT, we do not anticipate that combination toxicity should be greater than the known toxicities of the individual treatment modalities. However, this will need to be further assessed in future early phase clinical trials of the combination. **We have now included further discussion of the dosing of AZD5363 used in our work, increasing the therapeutic ratio and how this might translate into future combination clinical trial design (page 16, para 3)**.

Referee #2:

1. For the analysis of endothelial cells with ADZ5363 and IR the authors should consider constructing an isobologram to assess if there is an additive or synergistic effect.

We have previously demonstrated that treating endothelial cells with AZD5363 alone inhibits the proliferation of vascular endothelial cells, using a BrdU assay. In light of the reviewer's insightful comments we have conducted some further experiments to investigate whether there is a greater than additive effect from AZD5363 on the proliferation of HUVEC cells when treated with 6 Gy irradiation. **These additional experiments have been included (Extended View Figure EV3, and in the manuscript page 12, para 1)** and demonstrate that there was no additive increase in cell killing of the HUVEC endothelial cells regardless of whether AZDD5363 was given before and after the RT. Given this lack of additive cell kill we have not considered drawing an isobologram.

2. While the data on ADZ5363 on tumor cells shows that it inhibits Akt in tumor cells alone, it is unclear if this compound with IR is still effective as no western blots were provided.

As requested, we have now included a western blot demonstrating the effect of AZD5363 on irradiated FaDu cells (**Figure 1G and manuscript page 6, para 1**).

3. Is it possible that a subpopulation of tumor cells with IR+ADZ5363 undergo senescence, which would explain the decrease in VEGF?

The referee makes an interesting point and whilst it is possible that a subpopulation of tumour cells undergo senescence with the combination of RT+AZD5363, it would appear that senescence is not required in order to see a decrease in VEGF. The VEGF data depicted in Figure 5 demonstrates Human VEGF is reduced in FaDu tumours from mice treated with AZD5363 or vehicle alone. In this model, AZD5363 has no significant effect on tumour growth at the dose used (50 mg/kg BD), and our interpretation is that significant cellular senescence is unlikely to have occurred. In addition, in Figure 6E we demonstrate that Human Ki67 is not reduced in FaDu tumours with treatment with AZD5363 alone. This material is from the same tumours used to generate the VEGF data, again indicating senescence is not required for a reduction in VEGF. **A comment on this has now been made in the paper (page 14, para 1).**

4. The Figure S1 legend does not reflect the text on page 13 top paragraph.

We wish to thank the referee was noticing this error and have made the correction to the Figure reference.

5. It was not clear in Figure S5A and B if there was IR treatment alone. It was not clearly marked in the Figure.

We wish to clarify that there is no data for an RT alone treated mouse. Our intention was to compare combination treatment with RT alone, however the tumour on the RT alone treated mouse rapidly outgrew the tumour window necessitating culling of that animal only 3 days after treatment began. As such, this experiment was hypothesis generating, allowing us to focus our studies on the tumour vasculature to the 7 days post RT time point, using a different method of analysis; CD31 staining (immunohistochemistry). This has been made clearer in the figure legend.

6. In Figure 6 the data shows CD31 staining for vessels but it is unclear if the analysis was to measure all vessels. IT might be important to distinguish between mature vessels versus immature vessels.

In our analysis of blood vessels we felt it was important to consider the effect of AZD5363 post radiotherapy on all vessels, both mature and immature. As such we chose to use CD31 as it can be expected to stain all vessels (both mature and immature) rather than CD34, which tends to stain more immature vessels only. This was so as not to miss an effect on established, rather than just developing vasculature. **This has been clarified in the manuscript (page 11, para 2) and the mislabelling in Figure 4A has been corrected.**

7. The author suggesting that endothelial cells are most sensitive to ADZ5363 and IR. Could the author substantiate these finding by providing evidence that circulating endothelial progenitor cells in the animal are sensitive to the ADZ5363, IR or ADZ5363 and IR. This would provide evidence that source of the endothelial cells are from the bone marrow versus the mature vasculature.

The discussion in our original submitted manuscript outlined our interpretation of our results that whilst AZD5363 reduces the proliferative rate of vascular endothelial cells, it was the effects of AZD5363 in the wider tumour microenvironment that resulted in enhanced tumour control after radiotherapy. We did not present data to suggest a direct effect on the radiosensitivity of vascular endothelial cells with the addition of AZD5363. However, following the helpful reviewers comments, we performed additional experiments and the data from these experiments is included (**supplementary Figure S7**) demonstrating that there is not a greater than additive effect on HUVEC cell proliferation when AZD5363 and radiation are combined, suggesting endothelial cells are not especially sensitive to the combination. Furthermore, in three previously published studies it has been demonstrated that myeloid derived cells significantly contribute to tumour vasculogenesis after radiation whereas endothelial progenitor cells have a limited role (Kioi et al, 2010; Purhonen et al, 2008; Shinde Patil et al, 2005). Whilst an *in vivo* experiment to consider the question of the source of the endothelial cells is interesting proposition, our hypothesis and data, along with previously published works mentioned, do not support conducting this experiment.

We hope that our revisions will satisfy the reviewers. We have also reformatted the manuscript in order to meet the requirements as specified in the author guidelines.

Banerji U, Ranson M, Schellens JH, Esaki T, Dean E, Zivi A, Van der Noll R, Stockman PK, Marotti M, Garrett MD (2013) Abstract LB-66: Results of two phase I multicenter trials of AZD5363, an inhibitor of AKT1, 2 and 3: Biomarker and early clinical evaluation in Western and Japanese patients with advanced solid tumors. *Cancer Research* 73

Kioi M, Vogel H, Schultz G, Hoffman RM, Harsh GR, Brown JM (2010) Inhibition of vasculogenesis, but not angiogenesis, prevents the recurrence of glioblastoma after irradiation in mice. *The Journal of clinical investigation* 120: 694-705

Lamoureux F, Thomas C, Crafter C, Kumano M, Zhang F, Davies BR, Gleave ME, Zoubeidi A (2013) Blocked Autophagy Using Lysosomotropic Agents Sensitizes Resistant Prostate Tumor Cells to the Novel Akt Inhibitor AZD5363. *Clinical Cancer Research* 19: 833-844

Purhonen S, Palm J, Rossi D, Kaskenpää N, Rajantie I, Ylä-Herttuala S, Alitalo K, Weissman IL, Salven P (2008) Bone marrow-derived circulating endothelial precursors do not contribute to vascular endothelium and are not needed for tumor growth. *Proceedings of the National Academy of Sciences* 105: 6620-6625

Shinde Patil VR, Friedrich EB, Wolley AE, Gerszten RE, Allport JR, Weissleder R (2005) Bone Marrow-Derived $\text{lin}^{-}\text{c-kit}^{+}\text{Sca-1}^{+}$ Stem Cells Do Not Contribute to Vasculogenesis in Lewis Lung Carcinoma. *Neoplasia* 7: 234-240

2nd Editorial Decision

20 September 2017

Thank you for the submission of your revised manuscript to EMBO Molecular Medicine.

Unfortunately, reviewer 1 was not available to re-evaluate your manuscript and I therefore asked reviewer 2 to do so on his/her behalf. As you will see reviewer 2 is supportive, although s/he does mention a remaining issue with respect to your response to reviewer 1. I would suggest that you discuss these final concerns, including appropriate statistical analysis.

I am therefore prepared to accept your manuscript for publication pending satisfactory compliance with the reviewer's final requests. Please also fulfill the following editorial requirements:

Please submit your revised manuscript within two weeks. I look forward to seeing a revised form of your manuscript as soon as possible.

***** Reviewer's comments *****

Referee #2 (Remarks for Author):

Reviewer 1: The authors were responsive to the previous concerns raised. However, the data related to ADZ5363 cell cycle appears to show differences before irradiation, if the cell cycle phases (s and G2) are examined closely. It does not appear to have had stats performed on the cell cycle data either.

Reviewer 2: Overall the authors responded to the points raised.

Many thanks for agreeing to accept our manuscript for publication pending satisfactory compliance with the reviewer's final requests.

Reviewer 1: The authors were responsive to the previous concerns raised. However, the data related to ADZ5363 cell cycle appears to show differences before irradiation, if the cell cycle phases (S and G2) are examined closely. It does not appear to have had stats performed on the cell cycle data either.

Although we appreciate that when looking at the figure there is the suggestion of possible differences, statistical analysis did not show any difference when AZD5363 treated cells were compared with their respective irradiated and mock-irradiated controls. The type of analysis performed was the Kruskal-Wallis test followed by Dunn's multiple comparisons test. We apologise for not having included this information and this has now been added on p24 on the manuscript. We have also added labels to the figure to indicate the comparisons made. We hope that these changes will meet with your approval.

Corresponding Author Name: Emma Searle
Journal Submitted to: EMBO Molecular Medicine
Manuscript Number: # EMM-2017-07767-V2